# Impact of integrated care on trends in the rate of emergency department visits among older persons in Stockholm County: an interrupted time series analysis

Megan Doheny ![ORCID],[1] Janne Agerholm,[2] Nicola Orsini ![ORCID],[1] Pär Schön,[2] Bo Burström[1]

[1]Global Public Health, Karolinska Institute, Stockholm, Sweden
[2]Aging Research Center, Karolinska, Stockholm, Sweden

**Correspondence to**
Ms Megan Doheny;
megan.doheny@ki.se

## ABSTRACT

**Objective** To investigate the association between the implementation of an integrated care (IC) system in Norrtälje municipality and changes in trends of the rate of emergency department (ED) visits.

**Design** Interrupted time series analysis from 2000 to 2015.

**Setting** Stockholm County.

**Participants** All inhabitants 65+ years in Stockholm County on 31 December of each study year.

**Intervention** IC was established by combining the funding, administration and delivery of health and social care for older persons in Norrtälje municipality, within Stockholm County.

**Outcome** Rates of hospital-based ED visits.

**Results** IC was associated with a decrease in the rate of ED visits (incidence rate ratio: 0.997, 95% CI 0.995 to 0.998) among inhabitants 65+ years in Norrtälje. However, the rate of ED visits remained higher in Norrtälje than the rest of Stockholm in the preintervention and postintervention periods. Stratified analyses showed that IC was associated with a decline in the trend of the rate of ED visits among those 65–79 years, the lowest income group and born outside of Sweden. However, there was no significant decrease in the trend among those 80+ years.

**Conclusion** The implementation of IC was associated with a modest change in the trend of ED visits in Norrtälje, though the rate of ED visits remained higher than in the rest of Stockholm. Changes in the composition of the population and contextual changes may have impacted our findings. Further research, using other outcome measures is needed to assess the impact of IC on healthcare utilisation.

## INTRODUCTION

The population of Sweden is rapidly ageing, those 80+ years represent the fastest growing group and are expected to double within 25 years.[1] However, coupled with a longer life expectancy, there is an increased likelihood of older persons experiencing long-term health problems and functional disabilities

**Strengths and limitations of this study**

► The study used a long time series of population-based data and could adjust for changes in composition of the population, using robust methods.
► The geographic setting of Norrtälje made it difficult to find an appropriate control community.
► Other contextual changes during the study period may have impacted the results.

associated with old age.[2] Older persons often have complex health and social care needs due to multimorbidity, functional and cognitive impairments and frailty, therefore, frequently need health and social care services from multiple providers.[3] Older persons can experience reduced quality of care and adverse health outcomes due to uncoordinated and fragmented delivery of care services. Hence, it is important that the health and social care systems adapt to the needs of an ageing population, who could benefit from better alignment between and within health and social care services.[4 5]

The Swedish health and social care system is tax-funded and universal for all citizens, however, it is highly decentralised. The responsibility of provision, management and financing of services has three levels of governance. At the national level, the government sets policy, aims and directives, through legislation and economic incentives, while the 21 regions are responsible for the provision of health and medical care, and the 290 municipalities provide the social care services (ie, home-help including both household and personal care services) needed by older persons.[6] Regions and municipalities are independent, as they both collect taxes to finance the majority of care services, set

BMJ

priorities and make decisions at the regional and local level.[6] Moreover, market-orientated reforms have been introduced in both health and social care which promote competition, freedom of choice and diversity of care providers, further contributing to fragmentation in the Swedish care system.[7 8] Furthermore, there has been a reduction in the number of municipal institutional care places and in the number of hospital beds, resulting in a change in how long-term care is provided. These trends have resulted in a growing number of older persons with complex care needs living in the community.[6 9]

Primary healthcare (PHC) is the basis of the Swedish healthcare system, where most patients with chronic diseases are treated, and includes home-health care services. PHC should also coordinate with and be a link to social services for older persons. However, other specialist services may be required. The fragmentation in the Swedish system has placed those with complex care needs in a vulnerable position, as the patients must be able to obtain pertinent information relating to the care they need and which provider they should seek care from.[8 9] This increases the risk for individuals to experience poor continuity and coordination of care between multiple providers and may result in an increased use of emergency department (ED) care to meet their needs.[10] The ED serves as an important link between PHC services and the hospital-based care.[11] The staff in EDs are responsible for stabilising, diagnosing and treating patients, and moreover, they act as gatekeepers determining whether an acute patient will be admitted into inpatient care or organising and negotiating their discharge into the community.[12] However, those with complex needs could be better met through coordinated care with long-term support, rather than in an acute and episodic care setting.[13]

## Integrated care

Integrated care (IC) is an approach to organising and delivering care which could potentially provide care with greater efficiency and improve how patients receive and experience care.[14] In theory, IC might be effective at overcoming fragmentation of care.[14–18] The integration of care involves both vertical and horizontal processes. Vertical integration aims to bridge the gaps between different care services and providers that operate at different levels, that

is, from primary, secondary to tertiary care. Horizontal integration refers to when different care organisations combine their efforts and resources to develop a seamless system for delivering care that is, merging health and social care services and pooling resources.[13 14 16 17]

Healthcare systems are multilayered and complex, establishing IC involves consolidation within and between many different care providers.[14 16] The implementation of IC can be considered within three levels: the macro-level, that is, the system, the meso-level, that is, organisation and the micro-level, that is, the clinical.[15] The macro-level involves establishing structures to merge health and social care services across the continuum.[15 17 18] The meso-level involves the re-organisation of care services targeting those with greater needs. Finally, the micro-level focuses on the care required by individual users, to facilitate the coordination of care across the spectrum of providers, by involving the patient, their carers' and families' in the care decision to meet the goals of the patient.[17] An IC setting might better meet the needs of older persons as IC promotes better coordination and communication between professional groups which should lead to better continuity of care.[4 19]

## IC: the Norrtälje model

The Norrtälje model is an example of an IC initiative in Sweden. Figure 1 provides an overview of the gradual implementation of IC in Norrtälje. The initiative began when the local hospital was under threat of closure due to financial concerns. This led to a public appeal that gathered broad political support to save the local hospital. In 2006, IC was initiated as a shared project between Norrtälje municipality and Region Stockholm. A joint health and social care board with politicians from Region Stockholm and Norrtälje municipality was formed, responsible for the financial and organisation administration as well as the purchasing of care services from a jointly owned public company, Vårdbolaget TioHundra AB and other private care providers.[20–22]

The Norrtälje model was set-up to provide care to the entire population of the municipality. The intervention had its base at the hospital and set out to join the efforts of health and social care services, through aligning medical documentation, care planning, rehabilitation, preventative care, home-help, home-health

| | 2000 | 2001 | 2002 | 2003 | 2004 | 2005 | 2006 | 2007 | 2008 | 2009 | 2010 | 2011 | 2012 | 2013 | 2014 | 2015 | 2016 |
|---|---|---|---|---|---|---|---|---|---|---|---|---|---|---|---|---|---|
| Threat of closure of local hospital | | | ■ | | | | | | | | | | | | | | |
| Planning and preparation for IC | | | | ■ | ■ | ■ | | | | | | | | | | | |
| IC implemented – macro-structure and Tiohundra AB | | | | | | | ■ | | | | | | | | | | |
| Clinical level co-ordination of care | | | | | | | | ■ | ■ | ■ | ■ | ■ | | | | | |
| Shared information system | | | | | | | | | | | | ■ | ■ | ■ | | | |
| PHC Choice Reform in the rest of Stockholm County | | | | | | | | | ▨ | ▨ | ▨ | ▨ | ▨ | ▨ | ▨ | ▨ | ▨ |
| Social Care Choice Reform | | | | | | | | | | | ▨ | ▨ | ▨ | ▨ | ▨ | ▨ | ▨ |
| PHC Choice Reform in Norrtälje and the rest of Sweden | | | | | | | | | | | | ▨ | ▨ | ▨ | ▨ | ▨ | ▨ |
| IC project became permanent in Norrtälje | | | | | | | | | | | | | | | | | ■ |

**Figure 1** Overview of the timeline of the implementation of IC in Norrtälje. IC, integrated care; PHC, primary healthcare.

care and PHC services.[20] Care teams were created with specific purposes to facilitate inter-professional group meetings, for training and to improve service delivery. The key to this approach was the requirement that all providers must operate within the IC chain. The model facilitated the development of a variety of care paths (eg, dementia, stroke),[20] in addition to the national care plans for certain conditions already established in Sweden.[23] The Norrtälje Model used a shared approach to policy and financing in order to promote a better integration of resources and care staff, and further, provided a stimulus for the development of a shared information system to facilitate IC.[20 22] Our hypothesis was that the IC initiative in Norrtälje would facilitate better coordination of care and by doing so reduce the utilisation of ED visits among the inhabitants 65+ years in Norrtälje. This study aimed to investigate the potential association between the implementation of an IC system and the changes in the trends of ED visits in Norrtälje.

## METHODS

The study was designed with a quasi-experimental approach using repeated cross-sectional data, to assess the impact of IC on the utilisation of ED care by those 65+ years in Norrtälje compared with the rest of Stockholm County.[24] This study included all inhabitants 65+ years in Stockholm County on 31 December each year from 2000 to 2015. If an inhabitant died during the subsequent year, they were included in the study population for that year, if an inhabitant moved out of Stockholm Country during the study period they were excluded. Exposure to IC was measured by being a registered inhabitant of Norrtälje municipality. An interrupted time series (ITS) analysis was used to compare the rates of ED visits per quarter in Norrtälje before and after the implementation of IC in January 2006.[25–27] Subgroup analyses were done to study trends in specific groups (by sex, age, socioeconomic position, country of birth and living situation).

### Data sources

We used data from population-based registers covering the entire population of Stockholm County, obtaining data on health and healthcare utilisation from the Region Stockholm Healthcare Administrative Database (Vårdanalys databaserna (VAL)) which contains information on all registered outpatient and inpatient care visits financed by Region Stockholm. Data on ED visits were retrieved from the outpatient care register, and an ED visit was defined as an outpatient care visit to a hospital-based ED.

Sociodemographic variables were obtained from the Longitudinal Integration Database for Social Insurance and Labour Market Studies, this database contains a collection of variables from various population registers linked individually through encrypted personal identity numbers. Sex was categorised into male and female. Age was calculated using the registered year of birth. Individual income

was used as a measure of socioeconomic position, grouped into categories based on the distribution of the net annual equivalised household income and then ranked into quintiles. Country of birth was dichotomised into Sweden or other. Living situation was measured by civil status and was dichotomised into cohabiting or living alone.

### Patient and public involvement

The management group of Tiohundra AB were involved in the development of the research question and selection of the outcome measures based on the data available, they were not involved in the design and execution of the study. The data used in this study is based on encrypted personal identity number so that individuals in the study population are not identifiable. We have the possibility to disseminate our findings to the management group and the staff at Tiohundra AB.

### Statistical analysis

An ITS analysis with a generalised linear model with a Poisson distribution was used to evaluate whether the postintervention trends in the rates of ED visits differed significantly from the preintervention trends adjusting for the underlying trend and seasonality.[26] The quarterly counts of ED visits were the dependent variable, the independent variables included in the model were a dummy variable distinguishing between the preintervention=0, and the postintervention=1 period, and a variable counting the number of quarters that elapsed since the start of the study. This variable counting the elapsed time allows us to estimate the underlying trend of the outcome. We hypothesised that there would be a gradual trend change due to the intervention, so an interaction term between elapsed time and the intervention was included in the model. Additionally, we included a dummy variable distinguishing the preintervention and postintervention period to estimate the level change immediately following the implementation of IC in Norrtälje. Plots of the residuals and the partial autocorrelation function of the residuals were assessed to ensure that the adjustment for seasonal effects and autocorrelation were adequate.[25–27] The age-standardised rates of ED visits were used to adjust for the changes in the population over time. We had a total of 64 quarters as units of time in the study that were divided into two segments according to the intervention implementation. There were 25 quarters before, and 36 quarters after the implementation of IC. Stratified analysis was performed to investigate differential effects among sociodemographic groups which could potentially benefit from an IC system including, those 80+ years, in lowest income group, living alone and born outside of Sweden. We assessed trends in the rate of ED visits among the inhabitants 65+ years and the relevant subgroups in the rest of Stockholm County, to compare how the trends changed during the study period.

## RESULTS

In Norrtälje, the composition of the inhabitants 65+ years changed over time, the number of inhabitants increased

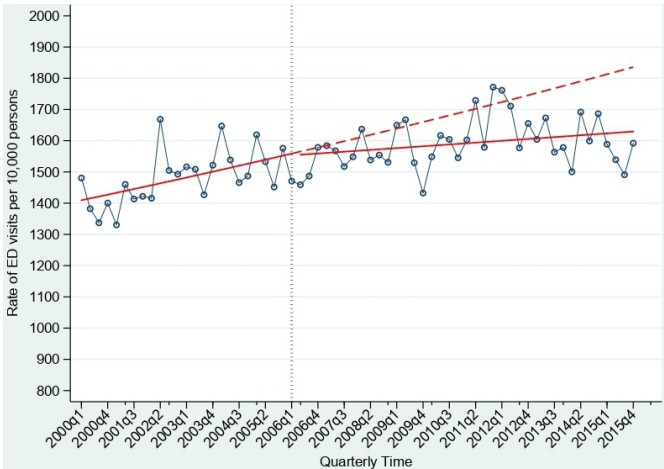

**Figure 2** The trend in the rate of ED visits among all inhabitants 65+ years in Norrtälje from 2000q1 to 2015q4. The circles represent the observed rate of ED visits per 10000 persons, the vertical line represents the implementation of integrated care (IC) (postintervention trend from 2006q1). The solid line represents the temporal trend, and the step at the postintervention period is the estimated effect and the dashed red line represents the counterfactual trend if IC was not implemented in Norrtälje. ED, emergency department.

from n=10 249 in 2000 to n=16 011 in 2015. The mean age decreased from 75.7 in 2000 to 74.8 years in 2015. Before the implementation of IC there was a larger proportion of women, while after the intervention the proportion of men 65+ increased by 3%. The proportion of inhabitants 65+ in the lowest income group decreased from 31% to 28%, while the proportion born outside of Sweden increased from 8.8% to 10.7%, the proportion of persons living alone remained constant throughout the study period (see online supplementary table S1).

In contrast, the population in the rest of Stockholm County, had a mean age 76.1 years in 2000 and 74.9 years in 2015, there was a higher proportion of inhabitants that were women, in the highest income group, born outside of Sweden and living alone (see online supplementary table S2). The inhabitants 65+ years in the rest of Stockholm County generally used lower levels of ED care compared with the inhabitants of Norrtälje; however, similarly to the inhabitants of Norrtälje the rate of ED visits increased between 2000 and 2015 (see online supplementary table S3).

### ITS analysis of the rate of ED visits among the inhabitants of Norrtälje

The relationship between the rate of ED visits and the implementation of IC among all inhabitants of Norrtälje 65+ years is demonstrated in figure 2 and table 1. There was an increasing trend in the rate of ED visits in the preintervention period. After adjusting for the underlying trends, we observed no substantial level change in the rate of ED visits (incidence rate ratio (IRR): 0.996, 95% CI 0.971 to 1.022). However, there was a significant trend change in the rate of ED visits (IRR: 0.997, 95% CI 0.995 to 0.998) in the postintervention period compared with the preintervention period. Figure 2 demonstrates the counterfactual trend based on the preintervention trend estimates of the quarterly rate of ED visits, which shows that in the absence of IC the rate of ED visits in Norrtälje would have had an increasing trend compared with the actual postintervention trend.

The stratified analyses showed a significant level change in the rate of ED visits among men (IRR: 0.959, 95% CI 0.937 to 0.983), immediately following the implementation of IC, but not among women (IRR: 1.036, 95% CI 1.009 to 1.064). We observed a significant trend change

**Table 1** The trend changes in the rate of ED visits associated with the implementation of IC in Norrtälje between 2000 and 2015

| | Rate of ED visits per quarter per 10000 population | | Age adjusted rate of ED visits per quarter per 10000 population | | Level change (immediately following the intervention) | | Trend change (after the intervention) | |
| --- | --- | --- | --- | --- | --- | --- | --- | --- |
| | Before | After | Before | After | IRR (95% CI) | P value | IRR (95% CI) | P value |
| All inhabitants | 1503.5 | 1562.7 | 1478.84 | 1584.17 | 0.996 (0.971 to 1.022) | 0.793 | 0.997 (0.995 to 0.998) | 0.000 |
| Males | 1627.0 | 1620.4 | 1674.5 | 1728.0 | 0.959 (0.937 to 0.983) | 0.001 | 0.997 (0.995 to 0.998) | 0.000 |
| Females | 1499.5 | 1510.6 | 1336.6 | 1493.6 | 1.036 (1.009 to 1.064) | 0.008 | 0.995 (0.994 to 0.998) | 0.000 |
| 65–79 years | 1218.1 | 1287.5 | 822.7 | 891.5 | 1.037 (1.003 to 1.073) | 0.034 | 0.995 (0.993 to 0.997) | 0.000 |
| 80+ years | 2145.8 | 2298.3 | 660.05 | 700.57 | 0.946 (0.911 to 0.983) | 0.005 | 0.998 (0.996 to 1.00) | 0.323 |
| Lowest income | 1582.2 | 1593.0 | 1449.6 | 1553.1 | 0.941 (0.917 to 0.965) | 0.000 | 0.996 (0.994 to 0.997) | 0.000 |
| Living alone | 1650.3 | 1764.4 | 1538.2 | 1676.6 | 0.987 (0.963 to 1.011) | 0.301 | 0.996 (0.995 to 0.998) | 0.000 |
| Born outside of Sweden | 1452.4 | 1683.0 | 1591.5 | 1748.0 | 0.872 (0.852 to 0.894) | 0.000 | 0.991 (0.989 to 0.992) | 0.000 |

The step change the level change immediately after the implementation of IC.
The trend change in the rate of ED visits in the postintervention period.
ED, emergency department; IC, integrated care; IRR, incidence rate ratio.

in the rate of ED visits for both men (IRR: 0.997, 95% CI 0.995 to 0.998) and women (IRR: 0.995, 95% CI 0.994 to 0.998) in the postintervention period compared with the preintervention period (see online supplementary figure S1 and S2). There was a decrease in the level of the rate of ED visits among those 80+ years (IRR: 0.946, 95% CI 0.911 to 0.983), however, this level change was not observed among those 65–79 years (IRR: 1.009, 95% CI 0.983 to 1.053). Furthermore, there was a significant decreasing trend change observed among those 65–79 years (IRR: 0.995, 95% CI 0.993 to 0.997) in the postintervention period compared with the preintervention period. However, there was no significant trend change in the rate of ED visits observed among those 80+ years (IRR: 0.998, 95% CI 0.997 to 1.001) in the postintervention period compared with the preintervention period (online supplementary figure S3 and S4).

There was a significant level change observed among those born outside of Sweden (IRR: 0.939, 95% CI 0.987 to 0.996); however, this decrease was not sustained throughout the postintervention period. Among those in the lowest income group we also found a significant level change (IRR: 0.916, 95% CI 0.878 to 0.954) as well as a significant trend change in the postintervention period. No significant level change was observed among those inhabitants living alone in Norrtälje. However, the rate of ED visits was no longer increasing in the postintervention period compared with the preintervention period (see online supplementary figures S5 and S7). A sensitivity analysis was performed on all inhabitants 65+ years using a shorter timespan 2000–2008. The trend of the rate of ED visits followed the same direction, which was observed in figure 2; however, the changes were not significant (see online supplementary figure S8).

### ITS analysis of the rate of ED visits among the inhabitants 65+ in rest of Stockholm County

The trend in the rate of ED visits among the inhabitants of the rest of Stockholm County differed from those in Norrtälje municipality in the preintervention period and it was therefore not possible to do a comparative interrupted time series analysis. The trends from the rest of Stockholm County are depicted in figure 3. From 2000q1 to 2005q4 there was a downward trend in the rate of ED visits. There was an increase in the level of ED visits; however, the trend between 2006q1 and 2015q4 was not significantly different from the trend in the preintervention period.

### DISCUSSION

Our findings suggest that the implementation of IC was associated with a decrease in the trend of ED visits in the postintervention period compared with the preintervention among all those 65+ years. However, the utilisation of ED care increased slightly among inhabitants in Norrtälje from the start of the preintervention period to the end of the postintervention period. Compared with the rest

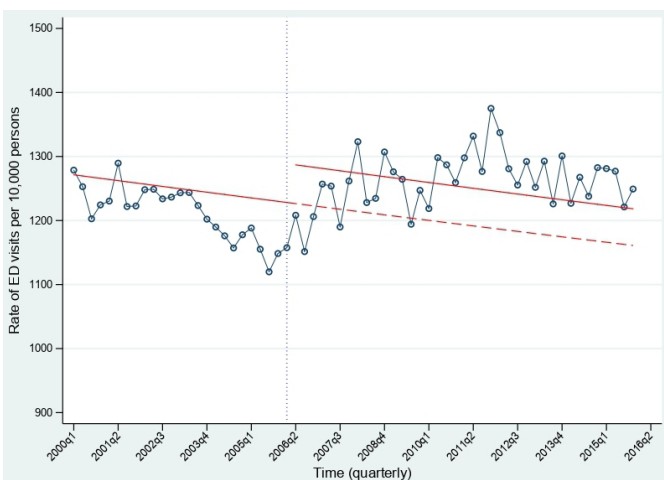

**Figure 3** The trend in the rate of ED visits among the inhabitants 65+ years in the rest of Stockholm County from 2000q1 to 2015q4. The circles represent the observed rate of ED visits per 10 000 persons, the vertical line represents the point where integrated care (IC) was implemented in Norrtälje. The solid line represents the temporal trend, and the step at the postintervention period is the estimated effect and the dashed red line represents the counterfactual trend if IC was not implemented in Norrtälje. ED, emergency department.

of Stockholm County the inhabitants of Norrtälje generally used more ED care. Additionally, the counterfactual demonstrated that the trend in the rate of ED visits would have increased if IC was not introduced. Our findings from the stratified analyses suggest that the rate of ED visits decreased among men, those 65–79 years, those in the lowest income group and those born outside of Sweden, after the introduction of IC. However, no significant change in the trend was observed among the inhabitants 80+ years. During the study period the demographic composition changed and number of inhabitants 65+ years increased in Norrtälje.

Previous studies investigating the Norrtälje Model, have been very positive to the effects of the integration of health and social care in Norrtälje.[20 22] These previous evaluations have been predominantly process evaluations interviewing stakeholders such as healthcare professionals, managers, politicians and patients.[21 22] Due to the geographic location of Norrtälje, the hospital is the primary source of care in the area, so inhabitants may have a habit of going to the hospital when in need of care. Furthermore, the impetus for establishing IC was to save the local hospital. Thus, this care-seeking behaviour may persist even after the implementation of IC.[22] These factors might contribute to explain the modest changes in the rate of ED visits we observed after the implementation of IC.

The implementation of IC in Norrtälje was an initiative supported by local and regional authorities. However, throughout the postintervention period additional nationwide policies were introduced, which may have influenced the effect of the intervention. After 2006 there

were changes in ambitions towards increasing freedom of choice, competition and diversity. This resulted in a purchaser–provider split in Norrtälje, which in turn, resulted in an increase of providers not part of IC, that competed with IC services.[28 29] However, the fundamental ideas of IC were retained, that is, integrating essential care services provided in older people's home.[29 30] Norrtälje maintained its own choice model in home-healthcare services until 2010 when the choice reform in primary care was implemented nationally.[7 28 30] From 2010, the TioHundra (IC provider) was no longer the only main care provider in Norrtälje. New providers were competing with the TioHundra which created an obstacle in the cohesion and collaboration needed between health and social care providers when developing IC.[29 30] This may have impacted our findings.

In Norrtälje, there was a top-down approach to implementing IC which started by combining funding from Norrtälje Municipality and Region Stockholm and through a shared management board. However, further steps toward integration were gradual and encountered professional and technological hurdles.[20 22] There were differences in occupational cultures, professional boundaries and the independent roles of healthcare professionals.[22] This required further staff training and improvement in communication between and within the services, as there was no infrastructure for information sharing between health and social care practitioners. A shared information system was established in 2013 which enabled care practitioners to more easily coordinate the care across different services.[20–22] Previous studies have indicated that financial factors are often barriers to achieving IC,[31] and having shared funding is assumed to facilitate coordinated care. However, it has been reported that IC has not resulted in a sustained reduction in hospital use.[15]

Previous research, investigating what impact IC could have on the use of ED care has shown mixed results.[12 18] In Denmark, studies of IC initiatives observed no effect on the use of ED care,[32] similarly, in the UK, the results of a difference-in-difference study of IC pilot studies reported no significant difference on the use ED services,[33] while a study from Australia observed that IC reduced ED visits for persons 65+ years with a chronic obstructive pulmonary disease or asthma diagnosis.[34] Other research has indicated that certain features of IC such as case management and multidisciplinary teams can be effective at reducing ED visits among older persons.[4 5 35] However, within the Norrtälje model micro-level changes in the delivery of care were introduced gradually at a later stage of implementation.

### Strengths and limitations

One advantage of this study was the use of population-based register data over a long time period. We could thereby take into account demographic changes in the composition of the population occurring during the study period. However, we had difficulty in finding an area that was comparable and followed a similar preintervention trend to Norrtälje to strengthen our ability assess if the changes observed were due to the intervention or other factors.[25 27] There are many potential explanations for this limitation. In Norrtälje, the hospital has a salient role in providing care in the community and therefore, inhabitants use hospital-based care in a different way, as it is a rural area which does not have the same access to specialist care compared with those in the rest of Stockholm County. Therefore, the generalisability of these results might be limited.

Furthermore, an ITS is most effective when there is a clear-cut change at a specific point in time that is expected to have an immediate effect.[25] For this study, although the macro-structure facilitating IC was established in 2006, this was a complex intervention which took place at many levels and as such the implementation was gradual.

Our outcome is the aggregated number of ED visits per point in time; however, we cannot measure the reasons for the visits. Therefore, we cannot distinguish if there were more ED visits because of poor care coordination or more acute conditions. Furthermore, due to the role that ED care plays in the healthcare system there might be some floor effects in the rate of ED visits, as there will always be some need for ED care.

The quasi-experimental nature of this study is both a strength and a limitation.[24] This approach is a strength as we were able to assess the impact of this large-scale shift to IC from a macro perspective, which gave us the possibility to follow the population before and after the intervention. Furthermore, having population data made it possible to study the total population and identify subgroups which might have benefited from IC. The downside to the quasi-experimental approach is our inability to avoid the impact of external influences,[24] as we have a long follow-up which coincided with other health and social care reforms which could influence our outcome. Finally, within this study we have no measures of how IC changed how inhabitants experienced care, whether the patients perceived that they received better care or were able to obtain the right combination of health and social care services to meet their individual needs. Previous research has observed that healthcare professionals working in an IC system perceive the quality of their provided care to be better.[22 30]

### CONCLUSION

In summary, we found that the establishment of the IC initiative in Norrtälje was associated with a modest decrease in the trend of the rate of ED visits among the inhabitants of Norrtälje. Due to the gradual implementation of different aspects of IC (eg, staff training and shared information system) along with other policy changes in health and social care in the postintervention period, it was difficult to disentangle the effect of the IC from the other policy changes. Further research is needed to better

understand what impact IC could have on the utilisation of other health and social care services.

**Acknowledgements** A special thanks to the Equity and Health Policy research group at the department of Global Public Health at Karolinska Institutet and the Aging Research Center for helpful comments and suggestion of this paper.

**Author contributions** MD, JA, NO, PS and BB conceived of the idea for, and participated in the design and the coordination this study. MD performed the statistical analysis and drafted the manuscript. MD, JA, NO, PS and BB were all involved in the interpretation of results and the revisions to the manuscript drafts. All authors read and approved the manuscript.

**Funding** The research leading to these results was carried out as part of the Social Inequalities in Ageing (SIA) project, funded by NordForsk, project no. 74637 and the Integrated Care Project funded by the Swedish Research Council for Health, Working Life and Welfare (Forte) project no. 2017-02155. The funders of this research have had no direct role in the design, data collection, analysis or interpretation of data and in the writing of the manuscript.

**Competing interests** None declared.

**Patient consent for publication** Not applicable.

**Ethical approvals** Ethical approval was obtained for the use of population-based data. The research team applied for, and was granted ethical approval from the Regional Ethical Review Board in Stockholm, Sweden Dnr: 2016/299–31.

**Provenance and peer review** Not commissioned; externally peer reviewed.

**Data availability statement** All data relevant to the study are included in the article or uploaded as supplementary information.

**ORCID iDs**
Megan Doheny http://orcid.org/0000-0002-5640-1239
Nicola Orsini http://orcid.org/0000-0002-2210-5634

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
