## [Reviewer comments · BMJ Open]

ARTICLE DETAILS

TITLE (PROVISIONAL)	Impact of Integrated Care on trends in the rate of Emergency Department visits among older persons in Stockholm County: An interrupted time series analysis
AUTHORS	Doheny, Megan Marie; Agerholm, J; Orsini, Nicola; Schön, Pär; Burström, Bo

VERSION 1 – REVIEW

REVIEWER	Maria Raven UCSF School of Medicine USA
REVIEW RETURNED	08-Jan-2020

GENERAL COMMENTS	This paper was well written. It's generalizability is limited for a few reasons including the specific municipality in which the IC intervention occurred, and due to the long and now possibly outdated timespan of this analysis, which ended in 2015. The IC intervention itself is outlined in a vague manner but the reader is left wondering exactly what the IC accomplished. For example, how did it integrate services specifically (e.g. by location, through care management) and precisely what services were integrated within their system? The description on page 5 is extremely vague. The conclusions seem overstated. Yes, the changes in ED visit rates are technically statistically significant, but are they clinically significant? the authors should translate their IRR findings into lay terms and explain exactly what they mean regarding ED visit reductions. In the results, the authors don't report the number of observations in their study. I am wondering if the reason for the statistical significance is due to a very large number of observations that provided a lot of power for the analysis. The time frame from 2000-2015 is almost too long. There are so many temporal trends that could occur (the authors discuss demographic shifts, for example, but are there others?) that could have also impacted health services use when an analysis spans 15 years. The ITS analysis may not be appropriate here, because the IC intervention (which needs to be much more specifically described) took place over multiple years and was not a single distinct, time limited change that could be easily contained in the analysis as the
---

	"intervention". I would recommend a statistical reviewer to determine if different methods might be needed. The references need to be updated--there don't seem to be many studies with publication dates after 2017, and most are years earlier than that.
--	---

REVIEWER	Carolyn Hullick University of Newcastle, Australia
REVIEW RETURNED	18-Jan-2020

GENERAL COMMENTS	Thank you for the opportunity to review this manuscript. As the authors discuss, integrated care is of interest to governments and health systems internationally as they grapple with providing care to increasing numbers of frail older people with complex health and social care needs. The authors have evaluated the introduction of integrated care in a municipality in Sweden using an interrupted time series analysis of ED presentations for people over 65 years across 15 years. I think the consideration of the following issues would strengthen the manuscript.  1. Integrated care in Norrtälje. I would like to better understand the model of care that is being evaluated. Did residents enrol in the program? What assessments were undertaken? What services were provided? Who determined what services were required.? Was there a role for the General Practitioner? How was care coordinated? 2. Integrated care generally targets patients with complex health and social care needs. As the authors state in supplementary table S1, around 70% of older people did not use the emergency department at all. Given the age cut off was 65 years, most of these people would not need integrated care. The analysis attempts to identify this through sub-group analyses: age stratification, income, living situation but I wondered if there are other stratification that would better identify the population that needs the program. Who was exposed to integrated care? Perhaps this is because the Social Insurance and Labour Market studies data base was used that did not include health measures eg numbers of medications. ED presentations and hospital admissions would also be markers but are the dependent variable. 3. It appears that the model changed over 7 years that integrated care was implemented. Should there be a period that the analysis is excluded ie a wash in period that allows integrated care to be established as I am sure it was a complex process. I am confident the research team would have discussed this. I was also not sure how death was managed in the analysis. 4. There are some minor issues with English that need addressing. In more detail Abstract: Page 2, Line 27, I was not sure about the phrase "associated with a decrease in t he trend of ED visits for both sexes" please review.
---

	Introduction: Page 3, line 30. has three governance levels to replace “divided between three levels of governance”. Line 47, please review sentence “Additionally, long-term care.....” Line 57, please review first sentence. How does the primary care system support health and social care in Sweden? Do they have a coordination role or is this left with the individual and their family? Page 5, line 18, “the region of Stockholm” AS mentioned above, I would like to better understand the Norrtalje model from a patient and family perspective. What services are provided, who coordinates care for an individual. Methods: Page 6: how was permission to use the databases approved? Was there a requirement for ethics review? Perhaps it is a publically available database? This also needs referencing. Results: Page 7, paragraph 1, line 38, I don't know that I understand what was unique about Norrtalje. Why were the ED presentation rates higher than the rest of Stockholm? Why did the authors state that there were no suitable control municipalities? (Page 11, line 60) Page 8, lines 37 to 52. I am a little unsure about what these sentences mean. There was a decrease in the level change for over 80 year olds (before and after) but then it says there was no significant difference in the trend change for over 80 year olds. Discussion: Page 9, line 43, last sentence of paragraph. What do the authors mean by our findings might be reflecting that the process of IC were slow and that there were challenges during the process? Page 10, line 16. Please explain why the introduction of new providers created an obstacle in cohesion and collaboration. I suspect some my confusion is related to not understanding what the Norrtalje integrated model of care is. Line 52, what does competition between primary care clinics and home help provides mean? I am presuming that the Norrtalje model allowed primary care clinics to provide home care? Page 12, few primary care clinics. What does this mean? Norrtalje was less primary care than other parts of Stockholm? Hospitals provide primary Care? Conclusion: Page 13, I wonder if the results are modest because the analysis included everyone over 65 years rather than the population that needed integrated care.
--	--

REVIEWER	Claire Pearson Wayne State University Department of Emergency Medicine Associate Professor Ascension St. John
REVIEW RETURNED	28-Jan-2020

GENERAL COMMENTS	pg 14 has ethics and consent info, this is often seen within the methods section
--

	The title is looking at ED visits among older persons, however the objective only states changes in trends in ED visits. Since the focus is on older persons I would include this in the objective There is agreement between the objective and the methods used. The description of the methodology and results are clear.
--	---

VERSION 1 – AUTHOR RESPONSE

Reviewer 1 comment: This paper was well written. It's generalizability is limited for a few reasons including the specific municipality in which the IC intervention occurred, and due to the long and now possibly outdated timespan of this analysis, which ended in 2015.

- Reply: Yes, the generalisability of our findings is something we must consider, and it is important to highlight it more when we discuss our strengths and limitations

- Action: We have elaborated on the generalisability of our findings, in the Discussion, Strengths and Limitations, page 12, lines 9-22.

Reviewer 1 comment: The IC intervention itself is outlined in a vague manner but the reader is left wondering exactly what the IC accomplished. For example, how did it integrate services specifically (e.g. by location, through care management) and precisely what services were integrated within their system? The description on page 5 is extremely vague.

- Reply: We agree that our description was not adequate.

- Action: We have revised the whole description of the Norrtälje Model, in Introduction section page 5, under the heading "Integrated Care- The Norrtälje Model" page 5, line 11-58.

Reviewer 1 comment: The conclusions seem overstated. Yes, the changes in ED visit rates are technically statistically significant but are they clinically significant? the authors should translate their IRR findings into lay terms and explain exactly what they mean regarding ED visit reductions.

- Reply: Yes, we should exercise caution and not overstate our findings. In terms of clinical significance, the use of ED by those 65+ years was no longer increasing as it was prior to the implementation of IC. The trends demonstrated that the intervention was more effective in certain groups, but we agree that the effects were modest.

- Action: We have modified the discussion and the conclusion to reflect our findings.

Reviewer 1 comment: The time frame from 2000-2015 is almost too long. There are so many temporal trends that could occur (the authors discuss demographic shifts, for example, but are there others?) that could have also impacted health services use when an analysis spans 15 years.

- Reply: Yes, we do have a long follow-up period from 2000-2015 which allows us to have a better overview of the period before the Norrtälje Model was implemented and allows us to follow how the implementation changed the trends of the rate of ED visits afterward. However, this study uses a quasi-experimental approach, which limits our ability to control for the other changes which occurred during the post-intervention period.

- Action: We have done sensitivity analyses with a shorter timespan, the results were not significant. We have reported this in the "Results" section page 9, lines 35-39 and have added the corresponding figure to the supplementary data, see figure S8.

Reviewer 1 comment: In the results, the authors don't report the number of observations in their study. I am wondering if the reason for the statistical significance is due to a very large number of observations that provided a lot of power for the analysis

- Reply: My apologies for not making it clearer in the text regarding the sample size, as the reviewer indicated the sample size with regard to the number of inhabitants 65+ years is quite large. However, in the time series analysis, the unit of analysis is the number of time points, each representing the aggregated outcome, the number of ED visits divided by the number of inhabitants in the area per each quarter of the year.

- Action: The number of time points included in the analysis is reported in the main text under the heading "Statistical Analysis" on page 7 line 45-42, and the number of inhabitants living in Norrtälje

and in the rest of Stockholm for each year of follow-up is reported in the supplementary data on Table S1 and Table S2.

Reviewer 2 comment: Integrated care in Norrtälje. I would like to better understand the model of care that is being evaluated. Did residents enroll in the program? What assessments were undertaken? What services were provided? Who determined what services were required.? Was there a role for the General Practitioner? How was care coordinated?

- Reply: We agree that our description was not adequate. It was not necessary for residents to enroll in the intervention, it was open to all based on their needs, as assessed by the care team upon contact with health care services. For persons receiving care in the community it was the district nurse under the supervision of the general practitioner, who determined what services were required. For older persons hospitalized it is the physician in charge and a case manager also known as a patient responsible nurse who determines the services required and navigates the transition from hospital. Further detail on how care was coordinated is available in the introduction.

- Action: We have revised the whole description see Introduction section under the heading “Integrated Care- The Norrtälje Model”, page 5, line 11-58.

Reviewer 2 comment: Integrated care generally targets patients with complex health and social care needs. As the authors state in supplementary table S1, around 70% of older people did not use the emergency department at all. Given the age cut off was 65 years, most of these people would not need integrated care. The analysis attempts to identify this through sub-group analyses: age stratification, income, living situation but I wondered if there are other stratification that would better identify the population that needs the program. Who was exposed to integrated care? Perhaps this is because the Social Insurance and Labour Market studies data base was used that did not include health measures eg numbers of medications. ED presentations and hospital admissions would also be markers but are the dependent variable.

- Reply: Yes, integrated care generally is aimed at providing care to patients with complex health and social care needs. And yes, the majority of those 65+ years did not use ED care but the Norrtälje Model was designed to provide care for the entire population of the Norrtälje municipality, so we do operate under the assumption that all inhabitants 65+ years were exposed to integrated care.

Through the stratified analysis we did attempt to detect those potentially more vulnerable groups who might have benefited more so from integrated care, such as those 80+ years, low income group, living alone and born outside of Sweden.

- The Longitudinal Integration Database for Social Insurance and Labour Market Studies (LISA) only contains socio-economic and demographic variables such as age, sex, country of birth, living situation and income. We retrieved data on the ED visits from the Region Stockholm Healthcare Administrative database, this database contains all register visits to outpatient and inpatient care which was financed by the Region Stockholm.

Reviewer 2 comment: It appears that the model changed over 7 years that integrated care was implemented. Should there be a period that the analysis is excluded ie a wash in period that allows integrated care to be established as I am sure it was a complex process. I am confident the research team would have discussed this.

- Reply: Yes, we did consider a wash-in a period, however, there is no natural time point in which we can say that there is fully integrated care, as the model is constantly being worked on and built upon in Norrtälje. Even now after the project has been made permanent the managing board of Tiohundra AB seek to improve the integration of services and communication between health and social care staff involved in caring for older persons.

- Action: We have done sensitivity analyses with a shorter timespan, the results were not significant. We have reported this in the “Results” section page 9, lines 35-39 and have added the corresponding figure to the supplementary data, see figure S8.

Reviewer 2 comment: I was also not sure how death was managed in the analysis.

- Reply: Yes, we did include those who died in the study population. The study population included all inhabitants 65+ years on the 31st of December prior to each year of the study period. Those who died

during the following year were included in the study population and their visits to ED care were counted. Please see supplementary data Table S1 and Table S2, where the proportion of inhabitants that died each year are reported.

- Action: We have made clearer in the “Methods” section, page 6, line 9-16.

Reviewer 2 comment: Abstract: Page 2, Line 27, I was not sure about the phrase “associated with a decrease in the trend of ED visits for both sexes” please review.

- Action: The sentence has been changed, in the Abstract, under Results heading, page 2, line 27
- Stratified analyses showed that IC was associated with a change in the trend of the rate of ED visits among those 65-79 years, the lowest income group and born outside of Sweden.

Reviewer 2 comment: Introduction: Page 3, line 30. has three governance levels to replace “divided between three levels of governance”. Line 47, please review sentence “Additionally, long-term care.....”

- Action: Sentence has been modified under the “Introduction” page 3, line 28.
- The responsibility of provision, management and financing of services has three levels governance.
- Action: Sentence clarified under the heading “Introduction” section, page 3, line 47-54.

Furthermore, there has been a reduction in the number of municipal institutional care places and in the number of hospital beds, resulting in a change in how long-term care is provided. These trends have resulted in a growing number of older persons with complex care needs living in the community (6, 9).

Reviewer 2 comment: Introduction: Line 57, please review first sentence. How does the primary care system support health and social care in Sweden? Do they have a coordination role or is this left with the individual and their family?

- Reply: The role of primary care in Sweden has been better described, see Introduction section, page 3 line 56 continued on page 4 up to line 10.

- Action: Primary health care (PHC) is the basis of the Swedish health care system, and where most patients with chronic diseases are treated, and includes home-health care services. PHC should also coordinate with and be a link to social services for older people, though, other specialist services maybe required. The fragmentation in the Swedish system has placed those with complex care needs in a vulnerable position, as the patients must be able to obtain pertinent information relating to the care they need and which provider they should seek care from (8, 9).

Reviewer 2 comment: Page 5, line 18, “the region of Stockholm” AS mentioned above, I would like to better understand the Norrtälje model from a patient and family perspective. What services are provided, who coordinates care for an individual.

- Reply: The terms Stockholm County refers to the geographical area of Stockholm, while Region Stockholm is the name of the organizational body responsible for the provision of health and medical care.

- Action: We have revised the whole description see Introduction section under the heading “Integrated Care- The Norrtälje Model”, page 5, line 11-58.

Reviewer 2 comment: Methods. Page 6: how was permission to use the databases approved? Was there a requirement for ethics review? Perhaps it is a publicly available database? This also needs referencing.

- Reply: Yes, the research team obtained ethical approval to use the databases from the Regional Ethical Review Board in Stockholm. However, the databases are not publicly available.

- Action: A statement regarding “Ethical Approval” has been included at the end of the manuscript after the conclusion, before the reference list.

Reviewer 2 comment: Results. Page 7, paragraph 1, line 38, I don't know that I understand what was unique about Norrtälje. Why were the ED presentation rates higher than the rest of Stockholm? Why did the authors state that there were no suitable control municipalities? (Page 11, line 60)

- Reply: Norrtälje is area wise the largest municipality in Stockholm County and is sparsely populated It has less health care facilities compared to other areas of Stockholm County. In Norrtälje, the hospital for historical reasons has a salient role in providing care in the community and therefore,

inhabitants use hospital-based care in a different way, as it is a rural area which does not have the same access to specialist care compared to those in the rest of Stockholm County. This might contribute to the higher rates of ED visits.

- In Norrtälje, the rate of ED visits were higher in Norrtälje than the other municipalities we considered using as a control area, as a consequence of this differing pre-intervention trend it was difficult to find a control area which was comparable in the interrupted time series model.

- Action: We have included these points in the “Discussion” section see page 10-12.

Reviewer 2 comment: Page 8, lines 37 to 52. I am a little unsure about what these sentences mean. There was a decrease in the level change for over 80 year olds (before and after) but then it says there was no significant difference in the trend change for over 80 year olds.

- Reply: This sentence has been changed to clearer in the interpretation of the results.

- Action: See “Results” section, page 9, line 14-18.

Reviewer 2 comment: Page 9, line 43, last sentence of paragraph. What do the authors mean by our findings might be reflecting that the process of IC were slow and that there were challenges during the process?

- Reply: This sentence we were attempting to consider how our findings may be influenced by how the process of forming IC took place over several years, and during the time there were additional major changes to the health and social care system which changed the delivery and utilization of care.

- Action: We removed sentence from the revised manuscript, as we deemed it to be confusing for the reader.

Reviewer 2 comment: Page 10, line 16. Please explain why the introduction of new providers created an obstacle in cohesion and collaboration. I suspect some my confusion is related to not understanding what the Norrtälje integrated model of care is.

- Reply: The new providers created an obstacle to the IC because they were free to establish practices in Norrtälje but were not required to align their operations with IC, as a result they often competed with IC.

- Action: A sentence has been added to the “Discussion”, page 11, line 3-10.

Reviewer 2 comment: Line 52, what does competition between primary care clinics and home help provides mean? I am presuming that the Norrtälje model allowed primary care clinics to provide home care?

- Reply: As this was not clearly described we have revised the discussion concerning these reforms.

- Action: This paragraph has been removed from the revised manuscript.

Reviewer 2 comment: Page 12, few primary care clinics. What does this mean? Norrtälje was less primary care than other parts of Stockholm? Hospitals provide primary Care?

- Reply: As Norrtälje is largely a rural municipality it has fewer health care facilities compared to other areas in Stockholm County. The hospital has a central role in integrating care but primary care clinics are important within the care chain, providing also home-based health care.

Reviewer 2 comment: Conclusion: Page 13, I wonder if the results are modest because the analysis included everyone over 65 years rather than the population that needed integrated care.

- Reply: Yes, it is true that not all those 65+ years would need or benefit from receiving integrated care. We did attempt through stratified analysis to identify vulnerable groups who could have benefited from the change to integrated care. However, we were limited in our ability to measure need of health care. But the Norrtälje Model had set out with specific programs for older people.

Reviewer 3 comment: pg 14 has ethics and consent info, this is often seen within the methods section

- Reply: Yes, the research team obtained ethical approval to use the databases from the Regional Ethical Review Board in Stockholm. However, the databases are not publicly available.

- Action: A statement regarding “Ethical Approval” has been included at the end of the manuscript after the conclusion, before the reference list.

Reviewer 3 comment: The title is looking at ED visits among older persons, however the objective only states changes in trends in ED visits. Since the focus is on older persons I would include this in

the objective

- Action: Title has been changed to align better with the study aim

VERSION 2 – REVIEW

REVIEWER	Carolyn Hullick University of Newcastle, Australia
REVIEW RETURNED	07-Apr-2020
GENERAL COMMENTS	I am happy that the authors have addressed the concerns in the first review. Thank your for your detailed responses